# Insights into the novel *Enterococcus faecalis* phage: A comprehensive genome analysis

**Sahar Abed[1,2], Mohammad Sholeh[2], Mahshid Khazani Asforooshani[2], Morvarid Shafiei ●[2]\*, Abdolrazagh Hashemi Shahraki[3], Shaghayegh Nasr[1,4]**

**1** Department of Microbial Biotechnology, Faculty of Basic Sciences and Advanced Technologies in Biology, University of Science and Culture, Tehran, Iran, **2** Department of Bacteriology, Pasteur Institute of Iran, Tehran, Iran, **3** Division of Pulmonary, Critical Care and Sleep, College of Medicine-Jacksonville, University of Florida, Gainesville, Florida, United States of America, **4** Microorganisms Bank, Iranian Biological Resource Center (IBRC), ACECR, Tehran, Iran

\* dr.m.shafiei@pasteur.ac.ir

**Data Availability Statement:** The genom sequence are availble in the NCBI database (accession number: OR762047).

## Abstract

*Enterococcus faecalis*, a Gram-positive bacterium, poses a significant clinical challenge owing to its intrinsic resistance to a broad spectrum of antibiotics, warranting urgent exploration of innovative therapeutic strategies. This study investigated the viability of phage therapy as an alternative intervention for antibiotic-resistant *E. faecalis*, with a specific emphasis on the comprehensive genomic analysis of bacteriophage SAM-E.f 12. The investigation involved whole-genome sequencing of SAM-E.f 12 using Illumina technology, resulting in a robust dataset for detailed genomic characterization. Bioinformatics analyses were employed to predict genes and assign functional annotations. The bacteriophage SAM-E.f 12, which belongs to the *Siphoviridae* family, exhibited substantial potential, with a burst size of 5.7 PFU/infected cells and a latent period of 20 min. Host range determination experiments demonstrated its effectiveness against clinical *E. faecalis* strains, positioning SAM-E. f 12 as a precise therapeutic agent. Stability assays underscore resilience across diverse environmental conditions. This study provides a comprehensive understanding of SAM-E.f 12 genomic composition, lytic lifecycle parameters, and practical applications, particularly its efficacy in murine wound models. These results emphasize the promising role of phage therapy, specifically its targeted approach against antibiotic-resistant *E. faecalis* strains. The nuanced insights derived from this research will contribute to the ongoing pursuit of efficacious phage therapies and offer valuable implications for addressing the clinical challenges associated with *E. faecalis* infections.

## Introduction

*Enterococcus faecalis* presents a significant challenge due to its multidrug resistance, including resistance to vancomycin, which is often considered a last-resort antibiotic [1]. This bacterium is intrinsically resistant to certain antibiotics, further limiting treatment options [2]. Studies have demonstrated high levels of resistance in *E. faecalis* against various antibiotics, with some strains showing resistance to multiple drugs [3]. The presence of antibiotic resistance genes in

**Funding:** The author(s) received no specific funding for this work.

**Competing interests:** The authors have declared that no competing interests exist.

*E. faecalis* genomes leads to the rapid selection of resistant strains, complicating infection treatment [4]. Recent data has shown that *E. faecalis* isolates have exhibited resistance percentages exceeding 40% against key antibiotics such as tetracycline, vancomycin, linezolid, erythromycin, and ampicillin [5]. The rise in antibiotic resistance among Enterococcus species, particularly *E. faecalis*, underscores the urgent need for the development of new anti-infective therapies [6–8]. The surge in antibiotic-resistant *E. faecalis* strains has increased the urgency for innovative therapeutic approaches, prompting keen interest in phage-based interventions [9–11]. This burgeoning interest stems from the demonstrated efficacy of phage therapy (PT) in exerting lytic activity against diverse antibiotic-resistant *E. faecalis* isolates [12–14], thus providing a promising alternative treatment [13].

The pivotal role of *E. faecalis* phages in the pursuit of novel therapeutic strategies has emerged [12, 15]. These phages exhibit the potential to shield against lethal vancomycin-resistant *E. faecalis*, as evidenced by murine bacteremia models, and contribute to restoring gut microbiota equilibrium [16]. In delving into the rich landscape of *E. faecalis* phages, research has uncovered a repertoire of lytic bacteriophages, including vB_EfaS_PHB08, vB_EfsS_V583, LG1, and others. These phages exhibit a commendable ability to target and disrupt *E. faecalis* biofilms, paving the way for potential applications in treating biofilm-related infections and dental root canal treatments [17–20]. The efficacy demonstrated in inhibiting biofilm formation and eradicating planktonic *E. faecalis* cultures attests to the promising role of these phages in clinical scenarios [12, 20, 21].

Moreover, investigations into the synergistic effects of bacteriophages and antibiotics revealed a potential avenue for enhancing treatment outcomes against multidrug-resistant *E. faecalis* infections. This collaborative approach offers hope of addressing the persistent challenge of antibiotic resistance associated with *E. faecalis* [19, 20].

However, a significant gap in our understanding remains regarding the genetic diversity, evolutionary mechanisms, and therapeutic applicability of *E. faecalis* phages, particularly in the context of antibiotic resistance. This knowledge gap brings us to the central question of our study: How can comparative genomics advance our understanding and application of *E. faecalis* phages in combating antibiotic-resistant infections?

Our objective is to conduct a comprehensive genomic analysis of isolated lytic bacteriophages targeting *E. faecalis*. This analysis aims to elucidate their genetic composition, diversity, and therapeutic potential against antibiotic-resistant strains. Through this exploration, we aspire to bridge existing knowledge gaps and lay the foundation for innovative phage-based therapeutic strategies. By unraveling the genetic intricacies of *E. faecalis* phages, this research seeks to contribute valuable insights into infectious disease research and the fight against antibiotic resistance, thereby paving the way for the development of novel therapeutic approaches.

## Method

### Phage isolation procedure

A total of 60 phenotypically confirmed *E. faecalis* isolates were sourced from Resalat Hospital in Tehran. These isolated were used later to verify the host spectrum of the isolated phages. To verify their identity, a specific primer set was employed in the polymerase chain reaction (PCR) amplification process. The forward primer sequence (`ACTTATGTGACTAACTTAACC`) and reverse primer sequence (`TAATGGTGAATCTTGGTTTGG`) used were reported in a prior investigation, ensuring the accuracy of the identification process [22].

The raw sewage samples obtained from Resalat Hospital in Tehran; Iran used as source for phage. The sewage sample was centrifugation at 8000 rpm for 20 min to eliminate

bacterial cells and debris. The supernatant, filtered through a 0.22 μm pore size membrane, was stored at 4˚C. The host bacterium, vancomycin-resistant *E. faecalis* (VRE) was cultured in BHI broth at 37˚C until the logarithmic growth phase was reached. Equal volumes of VRE cultures and sewage filtrates were amalgamated, supplemented with 2× brain heart infusion (BHI) broth, and incubated for 24 hours at 37˚C. Subsequently, the resulting phage lysate underwent purification via three successive cycles of single-plaque selection and co-culturing. Verification of phage isolation was conducted using the double-layer agar (DLA) method.

The plate was incubated at 37˚C for 24 h, and clear plaques indicated a successful phage infection.

Three rounds of single-plaque isolation were performed for purification. Phage titration involved tenfold serial dilutions in SM buffer (100 mM NaCl, 8 mM MgSO4, 50 mM Tris (pH 7.5) and 0.002% gelatin (w/v)), and plaques were enumerated after incubation, expressing titration as plaque-forming units (PFU/ml).

## Electron microscopy analysis

The examination of the structural characteristics of the isolated phage particles was conducted through Transmission Electron Microscopy (TEM), following established protocols [22]. In brief, a volume of 10 μL of purified phage particles was applied to a carbon-coated copper grid for a duration of 3–5 minutes, followed by staining with 1% (w/v) uranyl acetate at a pH of 7. Microscopic images were acquired using a Zeiss LEO 906 TEM (Carl Zeiss LEO EM 906 E, Germany) operating at an accelerating voltage of 100 kV.

## Whole genome sequencing and phage characterization

Following the extraction of the phage genome using a commercial kit (DNA Pure, FAVOR-GEN, Iran), which adhered to the manufacturer's guidelines and underwent rigorous quality assessment measures, including optical density measurements, gel electrophoresis, and a PCR assay targeting bacterial 16S primers to confirm the absence of bacterial genome contamination [23], the phage genome was subjected to comprehensive whole-genome sequencing (WGS) on the Illumina HiSeq 2000 platform [24].

The initial step involved the preprocessing of paired-end sequence reads of 150 bp obtained from the Illumina HiSeq 2000 platform, using Genomics Workbench CLC 22 [25]. This preprocessing included the trimming of adapters, ambiguous nucleotides, and low-quality sequences, following standard parameters (i.e., bases less than 15 nt, maximum 2 ambiguous nucleotides, and Q score ≤5 was trimmed).

The subsequent assembly of reads employed default parameters with a word size of 15 nt [25]. Validation of the obtained contigs was performed through a nucleotide BLAST using Geneious version 22 [26]. The selected contigs were then mapped to viral sequences in the GenBank database, and their association with putative bacteriophages was analyzed using the NCBI database [27]. To further validate the results, clean reads were mapped to the viral genomes identified in the BLAST results.

The comprehensive annotation of the whole genome was conducted using the CLC Genomic Microbial Plugin with reference genome-based annotated blast tools [25]. A 95% similarity threshold and a default E value of 0.0001 were utilized for annotation purposes.

Additionally, the genome candid was utilized for whole-genome alignment with selected NCBI phages, facilitated by the Whole Genome Alignment plugin in the CLC Genomics Workbench [27].

## Phage characteristics

**One-step growth experiment.** The replication dynamics of the phage were investigated through a one-step growth experiment [28]. This experiment involved monitoring phage lytic development and provided insights into the growth properties of the phage [29]. Mid-exponential phase VRE was precipitated, resuspended in phage lysate, and incubated at 37˚C. Samples were collected at 5-minute intervals over a 2-hour period. The burst size, indicative of the number of progeny phages released per infected cell, was determined using the DLA method.

## Multiplicity of infection (MOI) determination

The MOI was assessed to ascertain the optimal ratio of phage particles to host bacteria during infection [30]. This critical parameter provides insights into the efficiency of phage infection and was determined by carefully titrating the phage concentration against a fixed bacterial density.

## Host range determination

The specificity of the isolated bacteriophages towards their hosts was evaluated using the Spot test method on 60 clinical isolates of *E. faecalis*. This process entailed the application of phage lysates onto bacterial cultures to observe clear zones, which signify lytic activity. Furthermore, the lytic potential of these phages was also examined against additional bacterial species, including *Enterococcus faecium*, *Pseudomonas aeruginosa*, *Escherichia coli*, and *Staphylococcus aureus*. These additional bacterial strains were sourced from the Pasteur Institute of Iran's bacterial repository, ensuring a thorough analysis of the phages' host range.

## Absorption time assay

To explore the initial phases of infection, an absorption time assay was conducted [18]. In this experiment, mid-exponential phase VRE was subjected to exposure to the phage, with samples collected at 1-minute intervals over a 10-minute duration to monitor unabsorbed phages. The assay employed a double-layer agar system, enabling the quantitative assessment of phage concentrations by absorbance measurements. The recorded PFU (Plaque-Forming Units) values were subsequently analyzed, shedding light on the kinetics of phage attachment and entry into host cells. This approach provided detailed insights into the early events of the infection process.

## Environmental stability assay

The effect of temperature on phage viability was evaluated by incubating the phage suspensions at different temperatures (-20, 4, 37, 50, 60, and 70˚C) for 1h. Survival rates were determined using plaque assays [18].

The effect of pH on phage viability was investigated by incubating phage suspensions at different pH values (2, 4, 7, 10, and 14) for 1 h at 37˚C. Survival rates were determined using plaque assays.

The impact of calcium and magnesium ions on phage adsorption rate was assessed by incubating VRE and phage lysates with or without divalent cations. Samples collected at regular intervals were titrated to determine the proportion of adsorbed phage.

The stability of the phages in salts (NaCl) was evaluated by incubating the phage suspension with 5%, 10%, and 15% (w/v) salts for 1 h at 37˚C [31]. Survival rates were determined based on log PFU/ml and compared to those of the control.

### Effect of phage disruption on biofilms

Microtiter plates were set up by adding 20 μL of OD~0.5 VRE (vancomycin-resistant Enterococci) to 230 μL of Brain Heart Infusion (BHI) broth. We established biofilms at 1, 3, and 5 days old without phage exposure and then subjected them to phage treatment as outlined in reference [32]. Post biofilm formation, we treated them with 10 μL of phage lysate at varying concentrations ($10^{10}$ PFU/mL, $10^8$ PFU/mL, and $10^6$ PFU/mL) for 24 hours at 37°C. We quantified the absorption of 2,3,5-Triphenyltetrazolium chloride (TTC) stain in the phage-treated biofilms using a previously described method [33]. The biofilm experiments were performed in triplicate for both the phage-treated and untreated groups.

### *In vivo* assay for phage efficiency

The therapeutic efficacy of bacteriophage intervention against VRE infections was evaluated in a BALB/c mouse wound model [34]. The study involved the use of female mice, aged 5–6 weeks, divided into three experimental groups: a negative control group, a positive control group, and a treatment group infected with *E. faecalis* and subsequently treated with bacteriophages. The bacteriophage treatment involved the administration of a MOI of 1, six hours post-infection. Prior to treatment, an essential purification step was implemented to ensure the safety and purity of the bacteriophage suspension used. This involved the removal of lipopolysaccharides (LPS) using the EndoTrap® Red endotoxin removal kit, a widely recognized method for its efficacy in LPS clearance [34]. The effectiveness of LPS removal was confirmed using the Limulus Amebocyte Lysate (LAL) assay, ensuring that the bacteriophage suspension was free from endotoxins and safe for therapeutic application. Following the LPS removal, the bacteriophage suspension underwent concentration and buffer exchange, and was subjected to sterility tests and phage titer measurements to confirm the viability of the phages for therapeutic use [34].

### Histopathological analysis

The passage describes the histopathological analysis method employed in the study, involving the careful preparation of wound tissues from euthanized mice. After fixation, embedding, and sectioning, the tissues underwent standardized H&E staining for visualization. Blinded histopathological evaluation, conducted by a professional, included the assessment of various tissue parameters such as fibroblast activity, epidermal thickness, capillary count, hair follicles, collagen growth, neutrophil infiltration, and macrophage infiltration. The study utilized a detailed scoring system, as previously outlined in reference [35], allowing for a quantitative comparison of histopathological changes among different experimental groups. This approach enhances the precision of the analysis, providing valuable insights into variations across diverse experimental conditions.

## Results

### Isolation and multiplication of effective bacteriophage against *e. faecalis*

The bacteriophage SAM-E.f 12 was specifically isolated to target *E. faecalis*. The spot assay results, as illustrated in Fig 1A, along with the DLA assay outcomes presented in Fig 1B, unequivocally affirmed its lytic activity. The Transmission Electron Microscopy analysis, as depicted in Fig 1C, conclusively identified SAM-E.f 12 as a member of the *Siphoviridae* family. Noteworthy, the one-step growth curves elucidated a succinct latent period of 20 minutes and an impressive burst size of 5.7 PFU per infected cell.

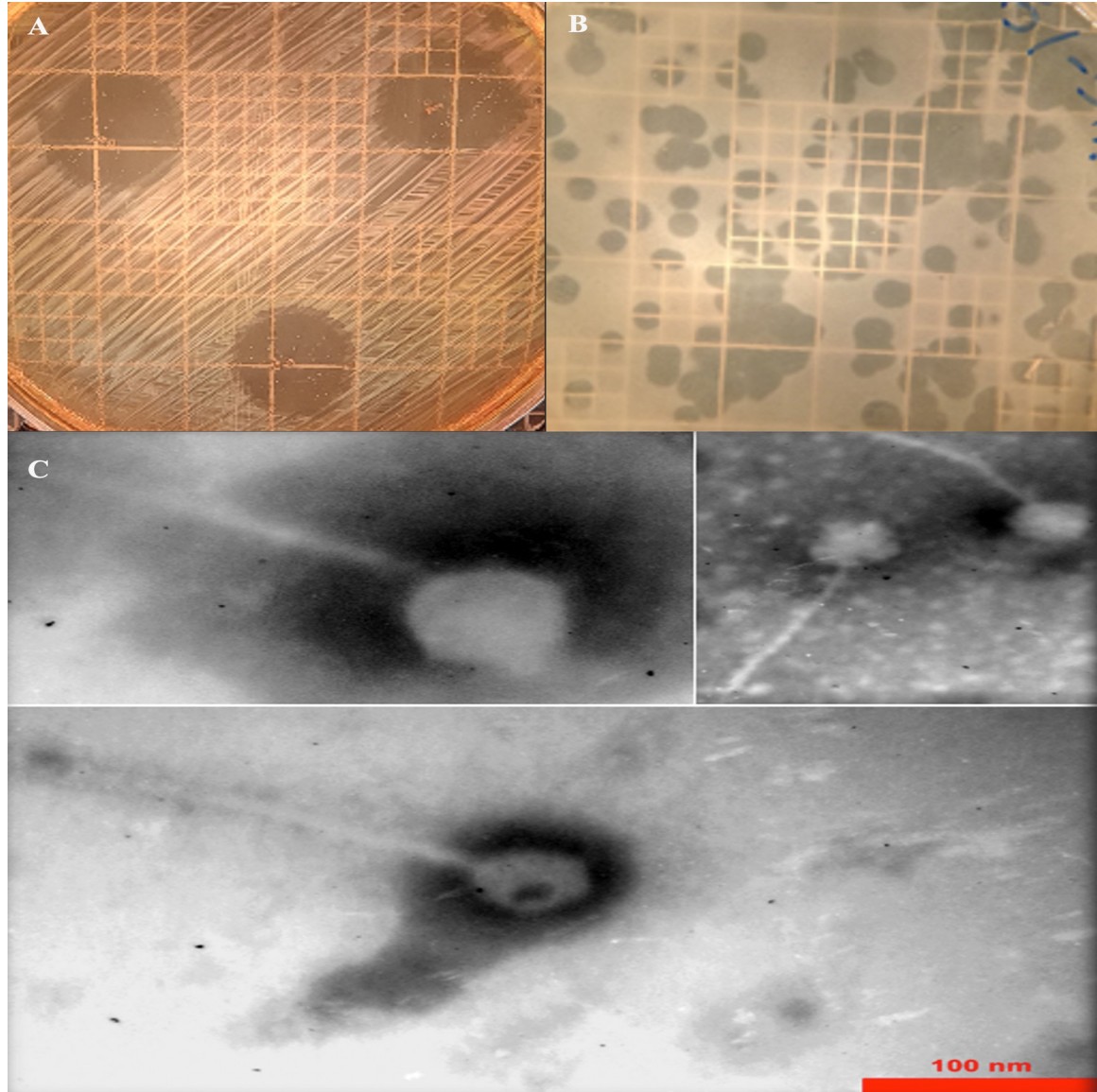

**Fig 1. Morphology of SAM-E.f 12 phage.** (A) Spot test showing plaques. (B) DLA method confirming lytic activity with abundant plaques and Bacteriophage Quantification. (C) Transmission electron micrograph of SAM-E.f 12 negatively stained with 2% w/v uranyl acetate, revealing a scale bar of 100 nm.

## WGS and bioinformatics analysis

A meticulous investigation of the SAM-E.f 12 genome revealed a robust contig coverage distribution, predominantly exceeding 100x, attesting to its high-quality assembly. Employing a specialized commercial column-based DNA extraction protocol ensured material purity without bacterial genomic contamination. WGS produced a comprehensive dataset of 1.5 Gb paired-end reads, designated GenBank accession number OR762047. Using Illumina technology, a robust dataset achieved an impressive 282x coverage of the SAM-E.f 12 genome. Rigorous quality assessment, precise trimming, and de novo assembly using metaS-PAdes resulted in a single contig spanning 36,626 bp, confirming the comprehensive coverage and precision.

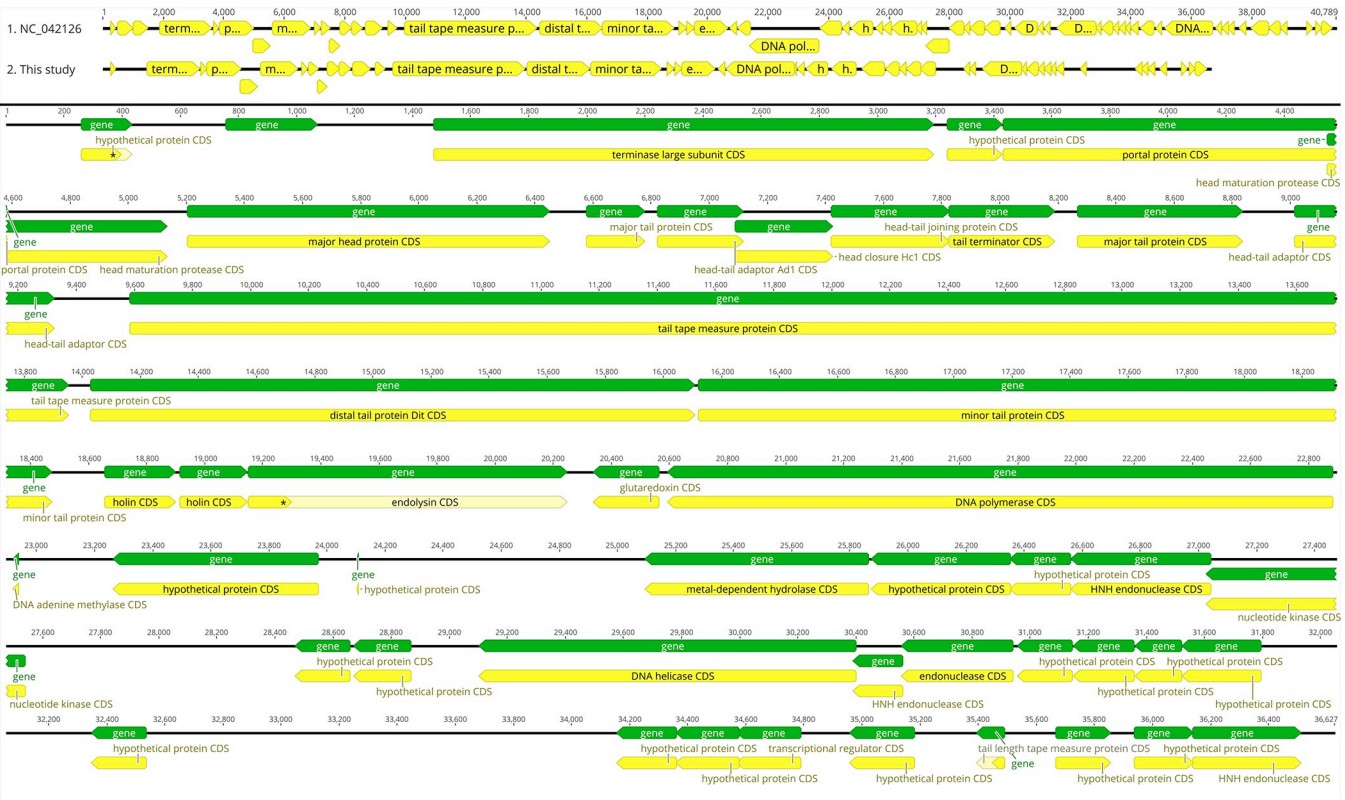

**Fig 2. Annotated genome map of SAM-E.f 12 bacteriophage.** The map displays sixty-one open reading frames (ORFs) represented as arrows, indicating the direction of transcription. Proposed modules are based on hypothetical functions inferred from bioinformatic analysis.

The annotated contiguous sequence was subjected to BLASTp analysis against the RefSeq viral database, enhancing our understanding of SAM-E.f 12's genomic composition and facilitating comparison with known sequences. Comparative genomics has placed SAM-E.f 12 within a sparse cluster in the *E. faecalis* phage population, inviting further exploration of its evolutionary dynamics and therapeutic potential. Comparative analysis with *Enterococcus* phage NC_0422125 revealed a query coverage of 98.3% and an APNI of 90.16%, emphasizing SAM-E.f 12's novelty and distinct genomic makeup (Fig 2).

This finding underscores the need for further research to elucidate SAM-E.f 12's specific characteristics. SAM-E.f 12's robust association with *Enterococcus* phage NC_0422125 in the APNI phylogenetic tree confirms its genetic similarity, supported by a compelling bootstrap value. The formation of a distinct clade involving SAM-E.f 12 and *Enterococcus* phage NC_0422125 signifies shared common ancestry, supporting the notion of genetic kinship among these phages (Fig 3). SAM-E.f 12's closer relation between *Enterococcus* phage NC_0422125 and its distinct divergence within the clade suggests dynamic evolutionary patterns among *Enterococcus* phages, thus contributing to our understanding of phage evolution and diversity.

SAM-E.f 12's alignment with *Enterococcus* phage NC_0422125, featuring a query coverage of 98.3% and an APNI of 90.16%, underscores its distinct genomic makeup. Despite minor misalignments, the accurate assembly reinforces the validity of the SAM-E.f 12 genome (Fig 3).

## Host range determination

The spot assays conducted on a diverse group of 60 clinical *E. faecalis* isolates provided crucial insights into the host specificity of SAM-E.f 12. In these assays, clear plaques were observed in

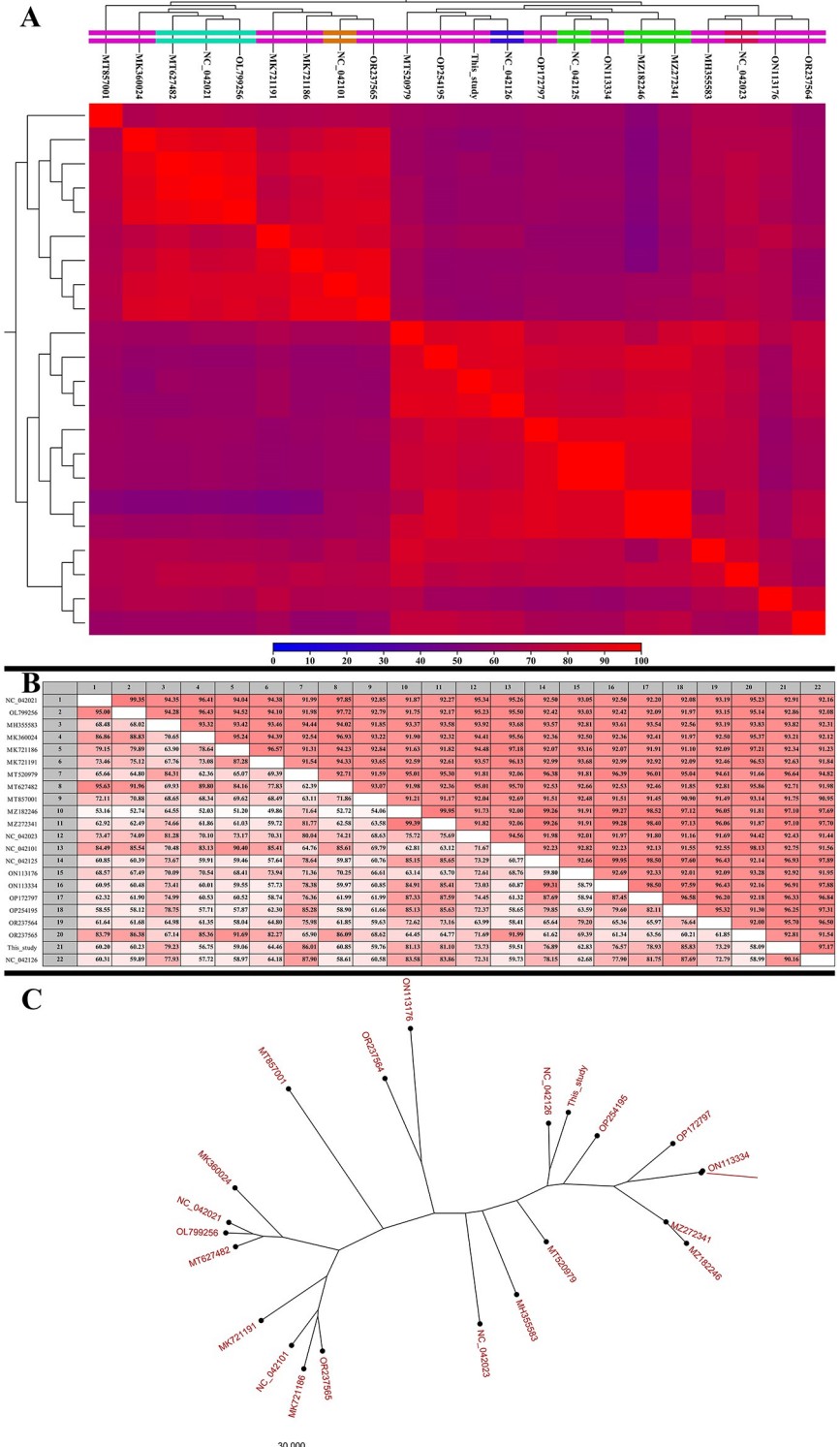

**Fig 3. Comparative analysis of phage genome alignment and AP heat map.** The Figure showcases the comparative analysis of SAM-E.f 12 genome alignment with Enterococcus phage NC_0422125. The alignment reveals a high query coverage of 98.3% and an Adjusted Percentage of Nucleotide Identity (APNI) of 90.16%. Despite minor misalignments, the accurate assembly of SAM-E.f 12's distinct genomic makeup is evident, highlighting its validity.

70% of the *E. faecalis* strains, demonstrating the phage's targeted effectiveness. Importantly, SAM-E.f 12 showed no lytic activity against *E. faecium*, *Pseudomonas aeruginosa*, *Escherichia coli*, and *Staphylococcus aureus*, confirming its specific action on select strains and illustrating its limited impact on a broader range of bacteria.

## Investigation of phage characteristics and stability to environmental factors

**Multiplicity of infection (MOI).** The determination of the MOI for SAM-E.f 12 phage is illustrated in Fig 4A, where a value of 0.1 was established. This critical metric provides insights into the ratio of phage particles to bacterial host cells, laying the foundation for subsequent analyses of SAM-E.f 12's lytic lifecycle parameters.

**One-step growth assays.** One-step assays were meticulously conducted to unravel SAM-E.f 12's lytic lifecycle parameters over a 240-minute post-infection period, as depicted in Fig 4B. The pivotal outcome of these assays was the determination of the average burst size, which was calculated as 5.7 PFU/infected cells. These quantitative data, when coupled with genomic analysis, significantly enhanced the comprehensive understanding of SAM-E.f 12's lytic lifecycle parameters. The integration of these findings contributes to the robust characterization of SAM-E.f 12 behavior during infection, providing valuable insights for both theoretical understanding and potential practical applications in phage therapy.

**Thermal stability.** Fig 4C illustrates SAM-E.f 12's thermal stability. The optimal lytic activity was recorded at 37˚C, which is consistent with the typical growth temperature of *E. faecalis*. The phage displayed detectable activity across a temperature range of -20˚C to 40˚C, with a significant decline beyond 50˚C. This suggests that SAM-E.f 12 possesses moderate thermal tolerance, which enhances its potential for applications in environments with variable temperatures.

## pH stability

The influence of pH on SAM-E.f 12's lytic activity, showcased in Fig 4D) revealed optimal performance within a pH range of 4–10, with a peak activity at pH 7. This versatility suggests SAM-E.f 12's effectiveness across diverse environmental pH conditions, which is a vital attribute for navigating varying pH gradients within the gastrointestinal tract.

## Salt tolerance

Fig 4E shows the response of the SAM-E.f 12 cells to salt stress. Although the number of phage particles decreased with increasing salt concentrations, sustained lytic activity was observed, indicating moderate salt tolerance. This characteristic enhances SAM-E.f 12's potential applicability in environments with elevated salinity such as certain food matrices.

## Adsorption rate

Fig 4F highlights SAM-E.f 12's adsorption rate of onto *E. faecalis*. Within 5 min, over 96% of the phage particles were efficiently absorbed by bacterial cells. This rapid adsorption is crucial for effective phage therapy, as it minimizes the timeframe available for the host to develop resistance.

## Investigation of phage impact on biofilm disruption

After staining and evaluating the optical absorption of the wells, we noted a significant decrease in absorption levels in most wells compared to the controls. A particular phage lysate,

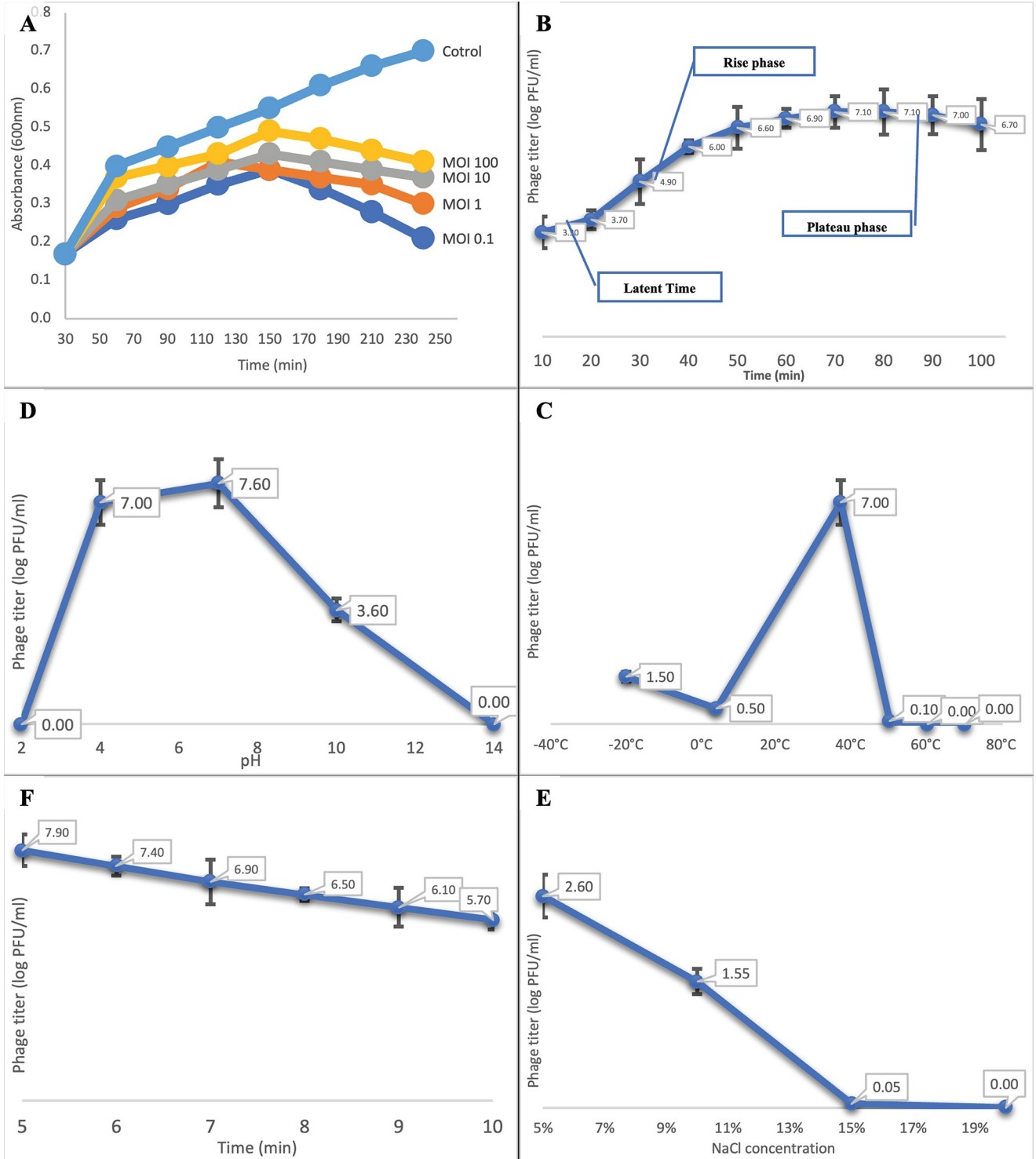

**Fig 4. Phage characteristics and environmental stability.** (A) MOI determination experiment of SAM-E.f 12 phage against *E. faecalis*. (B) One-step growth curve illustrating the replication dynamics of SAM-E.f 12 phage on *E. faecalis*. (C) Optimal lytic activity observed at 37˚C, with a temperature range of -20˚C to 40˚C. Lytic activity declined beyond 50˚C. (D) SAM-E.f 12 phage demonstrated optimal lytic activity between pH 4 and 10, peaking at pH 7. (E) Stability of SAM-E.f 12 phage under salt stress; the phage particle count reduced with increasing salt concentration, yet retaining lytic activity without reaching zero in tested samples. (F) Adsorption rate of SAM-E.f 12 phage on *E. faecalis*; more than 96% of phage particles were adsorbed by bacteria within 5 minutes.

at a concentration of $10^{10}$ PFU/mL, was markedly more effective in combating the biofilm. This was evident from the lowest absorption levels in these wells relative to other variants and control wells, indicating complete biofilm eradication. At a concentration of $10^8$ PFU/mL, the phage lysate was 100% effective against one-day-old biofilms, 80% effective against three-day-old biofilms, and 60% effective against five-day-old biofilms. When the phage lysate concentration was reduced to $10^6$ PFU/mL, it showed 100% effectiveness against one-day-old biofilms, 40% against three-day-old biofilms, and 20% against five-day-old biofilms.

### *In-vivo* efficacy testing

The in vivo assessment of phage therapy in the wound model of mice, as shown in the Fig 5, reveals striking results. Initially, all three groups had wounds of approximately 5mm. However, their healing trajectories diverged dramatically over time. The non-infected group, unsurprisingly, displayed the fastest healing, their wounds shrinking to 2mm by Day 7 and reaching full closure (0 mm) by Day 14. The infected control group showed sluggish progress, with wounds remaining around 4mm at Day 7 and only marginally reduced to 0.2mm by Day 14. Notably, the phage-treated group exhibited the most remarkable improvement. By Day 7, their wounds were significantly reduced to 1mm, and by Day 14, they achieved nearly complete healing with size of 0mm, mirroring the performance of the non-infected group. This striking outcome suggests that phage therapy significantly accelerates wound healing in infected mice, bringing their recovery rate closer to that of healthy control mice.

### Histopathological evaluation of wound healing outcomes in phage-treated

The histopathological evaluation of tissue parameters in different experimental groups reveals distinctive patterns of tissue responses, shedding light on the potential therapeutic effects of bacteriophage treatment. In comparison to the positive control group, characterized by intense fibroblast activity with a score of +3, significant collagen growth (+4), and pronounced neutrophil (+3) and macrophage infiltration (+4), the phage-treated group displayed a more controlled tissue response. The phage-treated group exhibited a moderate fibroblast activity (+1), moderate collagen growth (+2), and reduced neutrophil (+2) and macrophage infiltration (+3) compared to the positive control. Remarkably, the number of capillaries in the phage-treated group showed a substantial increase with a score of +3, indicating potential benefits for vascularization. Additionally, the hair follicles in the phage-treated group displayed notable improvement with a score of +3, suggesting positive effects on hair follicle regeneration. These findings suggest that bacteriophage treatment contributes to a more regulated and efficient wound healing process, mitigating excessive inflammation and promoting tissue repair. Comparisons with the negative control (no infection) and healthy mice groups underscore the need for further investigation to delineate the specific benefits and potential drawbacks of bacteriophage therapy in wound healing contexts. Overall, these detailed histopathological results provide a promising foundation for future in-depth studies exploring the therapeutic potential of bacteriophages in tissue repair and infection control (Table 1, and Fig 6).

### Discussion

*E. faecalis*, a Gram-positive bacterium, poses a formidable challenge in clinical settings due to its intrinsic resistance to a broad spectrum of antibiotics, including those vital for combatting bacterial infections [7, 8, 36]. This resistance has prompted its inclusion in the World Health Organization's priority pathogen list [37]. This resistance has not only placed *E. faecalis* on the World Health Organization's priority pathogen list but also highlighted the critical need for innovative therapeutic strategies to combat such resilient bacterial strains. Particularly

| Days | Negative Control | Positive Control | Treatment group |
|------|------------------|------------------|-----------------|

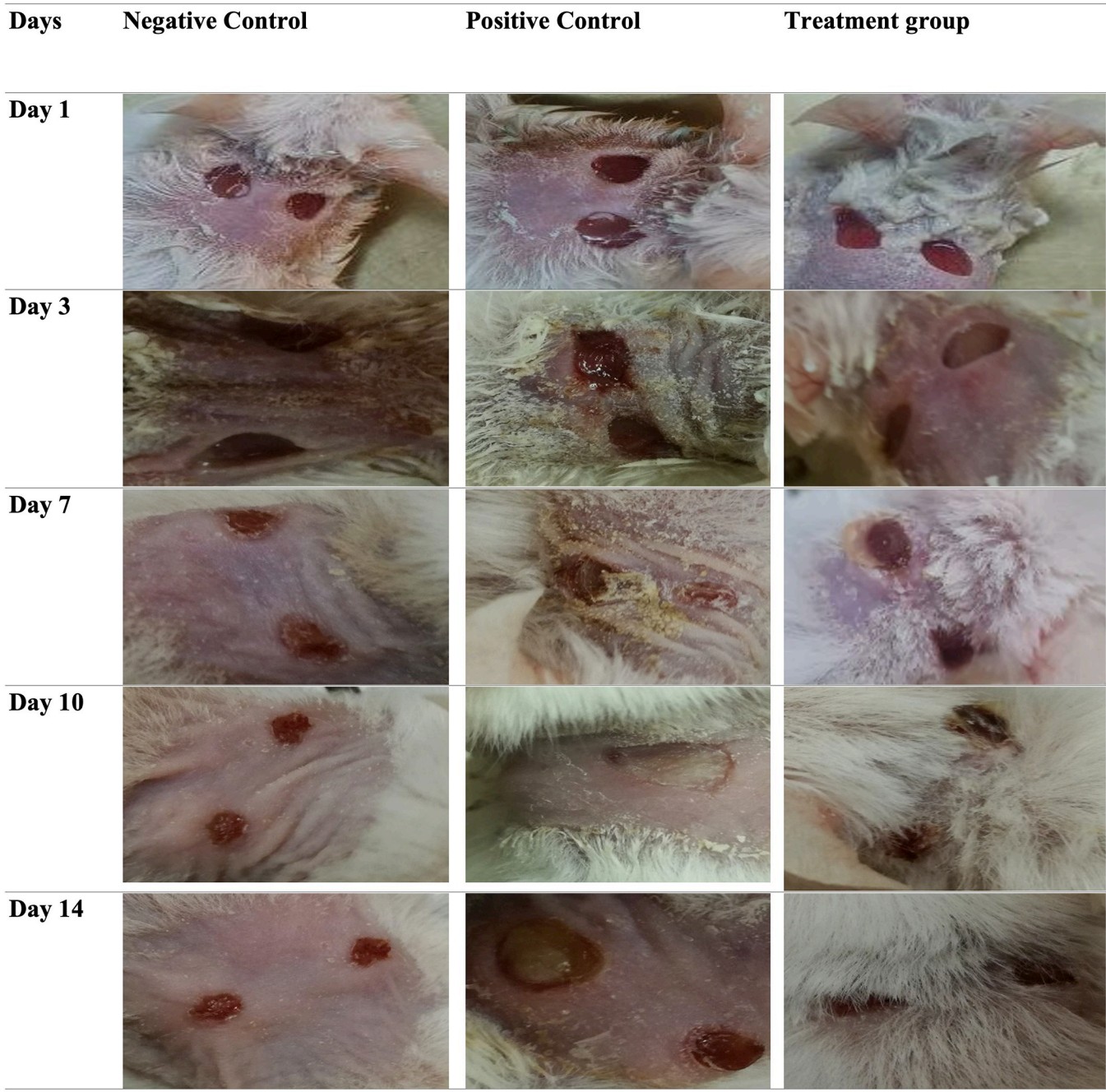

**Fig 5. Efficiency of phage therapy in *E. faecalis*-infected BALB/c mouse wounds.** The Fig demonstrates the notable enhancement of wound healing observed in the phage-treated group compared to the control group. The data provides evidence of the effectiveness of phage therapy in addressing *E. faecalis* infections in an in vivo BALB/c mouse wound model. The Fig presents the three groups: negative control (no bacterial infection), positive control (bacterial infection without phage treatment), and treatment group (infected with *E. faecalis* and treated with phage).

noteworthy is *E. faecalis*'s resistance to antibiotics like vancomycin, accentuating the exigency to explore alternative treatment modalities [2]. Phage therapy, characterized by a targeted approach, emerges as an advantageous strategy, especially in combating antibiotic-resistant *E. faecalis* strains, aligning with broader efforts to address antibiotic resistance through innovative antimicrobial exploration [2, 6].

**Table 1. Histopathological evaluation of tissue parameters in different experimental groups.**

| Categories | Phage-Treated | Positive Control | Negative Control | Healthy Mice |
|---|---|---|---|---|
| **Fibroblast Activity** | +1 | +3 | +3 | +1 |
| **Epidermal Thickness** | +1 | +1 | +2 | +1 |
| **Number of Capillaries** | +3 | +3 | +2 | +1 |
| **Hair Follicles** | +3 | +1 | +1 | +4 |
| **Collagen Growth** | +2 | +4 | +3 | +1 |
| **Neutrophil Infiltration** | +2 | +3 | +4 | +1 |
| **Macrophage Infiltration** | +3 | +4 | +3 | +1 |

Experimental Setup: The caption describes the different groups involved in the study. The "Phage-Treated" group refers to the experimental group that received bacteriophage treatment. The "Positive Control" group consists of injured mice with bacterial infection, serving as a comparison. The "Negative Control (No Infection)" group represents injured mice without bacterial infection. Lastly, the "Healthy Mice" group serves as the control group, consisting of mice without any injuries or infections.

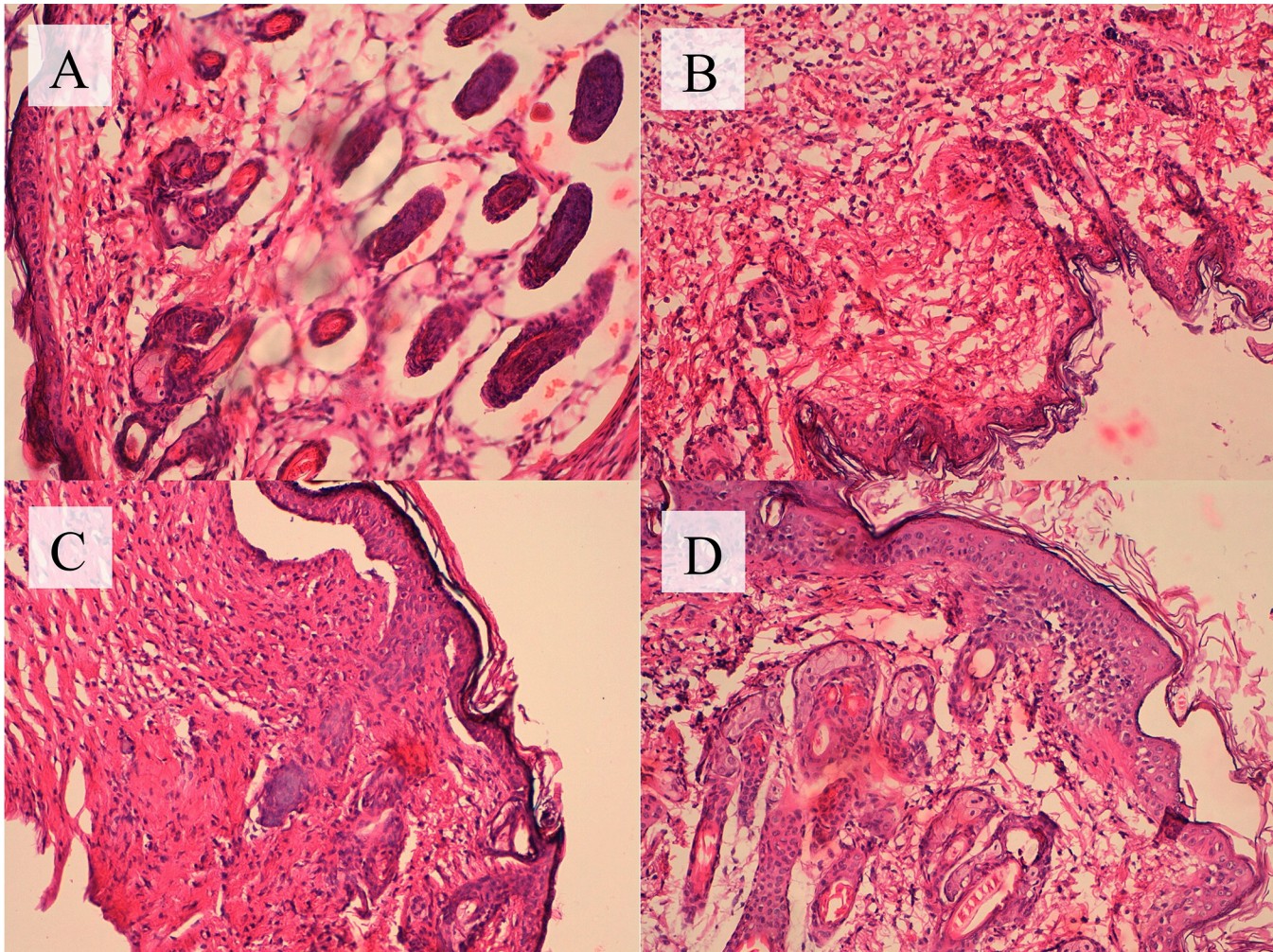

**Fig 6.** Histopathological Evaluation of Wound Healing Outcomes in Phage-Treated Group; (A): Phage-Treated; Experimental Group Treated with Bacteriophage, (B): Positive Control; Injured Mice with Bacterial Infection, (C): Negative Control (No Infection); Injured Mice without Bacterial Infection, (D): Healthy Mice; Control Group of Healthy Mice.

While systemic use of bacteriophages is hindered by limited understanding of their impact on human physiology, bacteriophage therapy presents distinct advantages in the context of wound infections, particularly those caused by *E. faecalis*. Previous studies have elucidated the lytic activity of *E. faecalis* bacteriophages against various Gram-positive pathogens, providing a foundation for their application in wound healing [19, 38]. Encouragingly, bacteriophages have shown efficacy in treating wounds infected by *Staphylococcus aureus*, chronic nonhealing wounds, and burn wounds infected by *Pseudomonas aeruginosa* [39]. These findings underscore the potential of bacteriophages in addressing specific infections, such as those caused by *E. faecalis*, and their significant role in wound healing. We acknowledge the pivotal role of previous studies in laying the groundwork for our research on bacteriophage therapy as a viable alternative to traditional antibiotics. These studies have demonstrated the effectiveness of bacteriophages in not only targeting *E. faecalis* but also in providing a basis for their application in treating various antibiotic-resistant bacterial infections [2, 6]. The selection of *E. faecalis* as our primary focus was driven by its alarming resistance to a wide array of antibiotics, including vancomycin, and its significant impact on healthcare systems worldwide.

The bacteriophage SAM-E.f 12, belonging to the *Siphoviridae* family, exhibits noteworthy lytic activity against *E. faecalis*, with a latent period of 20 minutes and a burst size of 5.7 PFU per infected cell [40]. Genomic analysis using BLASTp against the RefSeq viral database positions SAM-E.f 12 within a sparse cluster in the *E. faecalis* phage population. Comparative genomics with Enterococcus phage NC_0422125 highlights SAM-E.f 12's novelty, with a high query coverage of 98.3% and an APNI of 90.16%. The phylogenetic tree confirms SAM-E.f 12's genetic similarity to *Enterococcus* phage NC_0422125, suggesting shared common ancestry and dynamic evolutionary patterns among *Enterococcus* phages [41–43].

The bacteriophage SAM-E.f 12 demonstrated remarkable effectiveness against select isolates of clinical *E. faecalis*, evidenced by the formation of clear plaques in 70% of the 60 strains tested in the spot assay. This notable specificity, underscored by the lack of lytic activity against *E. faecium*, highlights SAM-E.f 12's targeted action, especially against vancomycin-resistant *E. faecalis* strains. Such specificity is crucial for addressing the growing challenge of antibiotic resistance and aligns with the findings of previous research, which have documented the efficacy of specific phages in selectively targeting *E. faecalis*. This alignment not only corroborates SAM-E.f 12's potential in phage therapy but also reinforces the importance of phage specificity in the development of targeted antibacterial treatments [44, 45].

The optical absorption results of this study significantly contribute to the understanding of SAM-E.f 12's biofilm disruption capabilities. At a concentration of $10^{10}$ PFU/mL, SAM-E.f 12 showed remarkable effectiveness in completely eliminating biofilms. This was evident from the significantly lower absorption levels observed compared to the controls and other wells. The concentration-dependent efficacy of SAM-E.f 12 was evident, with complete eradication of one-day-old biofilms at $10^8$ PFU/mL. The phage also displayed an 80% success rate against three-day-old biofilms and a 60% effectiveness against five-day-old biofilms at this concentration. However, when the concentration was reduced to $10^6$ PFU/mL, the effectiveness declined, especially against older biofilms. While it achieved 100% effectiveness against one-day-old biofilms, the success rates dropped to 40% and 20% for three-day and five-day-old biofilms, respectively. These findings indicate that the potency of SAM-E.f 12 in biofilm disruption is influenced by both the concentration of the phage and the age of the biofilms.

These findings emphasize the potential of SAM-E.f 12 as a therapeutic agent against biofilm-associated infections. The concentration-dependent efficacy underscores the importance of optimizing phage concentrations to achieve maximum therapeutic benefits, particularly when dealing with older and more resilient biofilms. Consequently, this study positions SAM-E.f 12 as a promising candidate for antibacterial treatments, especially in

cases where traditional antibiotics struggle to combat the persistent nature of biofilms. These results indicate the significant prospects of using SAM-E.f 12 as a targeted therapy for biofilm-related infections, highlighting its potential contribution to the field of antibacterial treatments [12, 46].

The results of this study underscore the therapeutic potential of SAM-E.f 12 in addressing biofilm-associated infections. The concentration-dependent efficacy highlights the need for optimizing phage concentrations to maximize therapeutic benefits, particularly when dealing with established and resilient biofilms. These findings position SAM-E.f 12 as a promising candidate for antibacterial treatments, especially in cases where traditional antibiotics face challenges in targeting biofilms. The study suggests that SAM-E.f 12 could serve as a targeted therapy for biofilm-related infections, offering promising prospects in the field of antibacterial treatments [47–49]. One-step assays reveal an average burst size of 5.7 PFU/infected cells over a 240-minute post-infection period, enhancing our understanding of SAM-E.f 12's behavior during infection [50]. These quantitative data, combined with genomic analysis, contribute to a robust characterization of SAM-E.f 12's lytic lifecycle parameters, offering valuable insights for both theoretical understanding and practical applications in phage therapy.

SAM-E.f 12's optimal lytic activity at 37°C, detectable across a broad temperature range (-20°C to 40°C), aligns with the physiological conditions of *E. faecalis* growth [51]. Moreover, optimal lytic activity within a pH range of 4–10, with peak activity at pH 7, and moderate salt tolerance further enhance SAM-E.f 12's potential applicability in diverse environmental conditions, such as the gastrointestinal tract and certain food matrices [52, 53]. The rapid adsorption of SAM-E.f 12 onto bacterial cells, a crucial factor for effective phage therapy, underscores its potential in minimizing the timeframe for the development of host resistance [54, 55].

In vivo assessment of SAM-E.f 12 phage therapy in a mouse wound model demonstrates accelerated wound healing, bringing recovery rates closer to healthy controls. This aligns with previous studies showcasing the efficacy of phage therapy in various animal models and bacterial infections [56, 57]. Histopathological evaluation reveals distinctive tissue responses, with the phage-treated group displaying a more controlled tissue response, increased capillaries, and improved hair follicles. These findings suggest that SAM-E.f 12 contributes to a regulated and efficient wound healing process, showcasing potential benefits for tissue repair and infection control [39, 58].

## Conclusion

In conclusion, the multifaceted exploration of SAM-E.f 12's characteristics positions it as a promising candidate for phage therapy against antibiotic-resistant *E. faecalis* infections. Its distinct genomic makeup, targeted lytic activity, and adaptive environmental features underscore its potential significance in addressing challenges posed by antibiotic-resistant strains and persistent infections. Further research, especially in in vivo settings and clinical trials, will be pivotal in substantiating its therapeutic efficacy and expanding its application in real-world scenarios.

## Acknowledgments

We extend our sincere thanks to Dr. Ms. Fatemeh Ashrafi at Resalat Hospital, Tehran, for her crucial support, Dr. Abozar Ghorbani for his collaboration in bioinformatics, and the Pasteur Institute of Iran for providing laboratory facilities. Their contributions have been invaluable to our study.

## Author Contributions

**Conceptualization:** Sahar Abed, Mohammad Sholeh, Abdolrazagh Hashemi Shahraki, Shaghayegh Nasr.

**Data curation:** Sahar Abed, Morvarid Shafiei.

**Formal analysis:** Sahar Abed, Mohammad Sholeh.

**Investigation:** Shaghayegh Nasr.

**Methodology:** Sahar Abed, Mohammad Sholeh, Morvarid Shafiei, Shaghayegh Nasr.

**Resources:** Mahshid Khazani Asforooshani.

**Software:** Sahar Abed.

**Validation:** Mahshid Khazani Asforooshani, Morvarid Shafiei.

**Visualization:** Sahar Abed, Morvarid Shafiei.

**Writing – original draft:** Sahar Abed, Mohammad Sholeh, Mahshid Khazani Asforooshani, Morvarid Shafiei.

**Writing – review & editing:** Mohammad Sholeh, Abdolrazagh Hashemi Shahraki.

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
