## [Decision Letter · Decision Letter 0]

21 Feb 2024

PONE-D-24-03766Insights into the Novel Enterococcus faecalis Phage: A Comprehensive Genome AnalysisPLOS ONE

Dear Dr. Shafiei,

Thank you for submitting your manuscript to PLOS ONE. After careful consideration, we feel that it has merit but does not fully meet PLOS ONE’s publication criteria as it currently stands. Therefore, we invite you to submit a revised version of the manuscript that addresses the points raised during the review process.

We look forward to receiving your revised manuscript.

Kind regards,

Vinod Kumar Yata, PhD

Academic Editor

PLOS ONE

Journal Requirements:

Additional Editor Comments:

The manuscript titled "Insights into the Novel Enterococcus faecalis Phage: A Comprehensive Genome Analysis" lays a robust foundation for a promising investigation into phage therapy against antibiotic-resistant E. faecalis.

Authors are requested to adhere to the "manuscript body formatting guidelines," with a specific emphasis on modifying the abstract in accordance with the provided guidelines. It is essential to ensure that the revised abstract aligns appropriately with the specified formatting requirements.

Authors are kindly urged to resubmit the manuscript after addressing the reviewers' comments. We appreciate your cooperation and anticipate the revised submission to meet the specified formatting guidelines and further enhance the overall quality of the manuscript.

Reviewers' comments:

Reviewer's Responses to Questions

**Comments to the Author**

1. Is the manuscript technically sound, and do the data support the conclusions?

Reviewer #1: Yes

Reviewer #2: Yes

2. Has the statistical analysis been performed appropriately and rigorously? 

Reviewer #1: Yes

Reviewer #2: Yes

3. Have the authors made all data underlying the findings in their manuscript fully available?

Reviewer #1: Yes

Reviewer #2: Yes

4. Is the manuscript presented in an intelligible fashion and written in standard English?

Reviewer #1: Yes

Reviewer #2: Yes

5. Review Comments to the Author

Reviewer #1: The overall work is good. Methods and materials results and discussions are represented clearly.

I encourage the authors to revise their manuscript and to resubmit it to the journal. This work is technically sound and isolation of phages is very difficult but isolation and charectarisation do it perfectly.

Reviewer #2: The research article “Insights into the Novel Enterococcus faecalis Phage: A Comprehensive Genome Analysis” by Abed et al., provides important insights on the bacteriophage SAM-E.f 12 genomic potentials. The abstract is well written however the introduction is poorly cited and the research question and context on importance of bacterial comparative genomics needs to be established. With the minor modifications I believe the manuscript can be accepted for publication in Plos One.

The specific comments are as follows.

1. Line 48: enlist the class of antibiotics bacteria E. faecalis known for with appropriate citations.

2. Line 49-51: Citation needed.

3. Methods results are well written and represented.

6. PLOS authors have the option to publish the peer review history of their article (what does this mean?). If published, this will include your full peer review and any attached files.

Reviewer #1: No

Reviewer #2: **Yes: **DSC

---

## [Author Response · Author response to Decision Letter 0]

8 Mar 2024

Reviewer #1: The overall work is good. Methods and materials results and discussions are represented clearly.

I encourage the authors to revise their manuscript and to resubmit it to the journal. This work is technically sound and isolation of phages is very difficult but isolation and characterization do it perfectly.

Response to reviewer 1: 

Thank you for your encouraging feedback and constructive comments on our manuscript. We are heartened by your recognition of the clarity of our methods, materials, results, and discussions, as well as your acknowledgment of the technical soundness of our work and the successful isolation and characterization of phages.

We will take your encouragement to revise and resubmit our manuscript as an opportunity to further refine our work. We appreciate the support and guidance provided by your feedback, which we believe will enhance the overall quality and impact of our study.

Sincerely,

Reviewer #2:

The research article “Insights into the Novel Enterococcus faecalis Phage: A Comprehensive Genome Analysis” by Abed et al., provides important insights on the bacteriophage SAM-E.f 12 genomic potentials. The abstract is well written however the introduction is poorly cited and the research question and context on importance of bacterial comparative genomics needs to be established. With the minor modifications I believe the manuscript can be accepted for publication in Plos One.

Response to reviewer: We have revised the introduction section according your suggestions.

The specific comments are as follows.

1. Line 48: enlist the class of antibiotics bacteria E. faecalis known for with appropriate citations.

We have updated Line 46-55 to include a list of the classes of antibiotics that Enterococcus faecalis is known to resist, accompanied by appropriate citations to ensure comprehensive support for our statements.

2. Line 49-51: Citation needed.

In all introduction section we have now incorporated the necessary citations to fully reference the background information provided in this section, thereby strengthening the manuscript's credibility and scholarly depth.

3. Methods results are well written and represented.

We are pleased to hear your positive feedback on the methods and results sections. We believe that the clarity and representation of these sections contribute significantly to the overall quality of our research.

We are confident that these revisions address your concerns and enhance the manuscript's suitability for publication in Plos One. We look forward to the possibility of our work being accepted and thank you again for your valuable insights.

Sincerely,

Reviewer #3:

Reviewer comment 1: The overall work is good. Methods and materials results and discussions are represented clearly. 

Dear Reviewer,

Thank you for your positive feedback and constructive comments regarding our manuscript. We are grateful for the encouragement to revise and resubmit our work, and we have meticulously addressed the specific concerns you raised to enhance the clarity and impact of our study. Below, we detail our responses and the amendments made to the manuscript in accordance with your suggestions.

Reviewer comment 2: I encourage the authors to revise their manuscript and to resubmit it to the journal.

Give the explanation regarding the following comments:

• In discussion part please mention supporting results of previous study.

Response to reviewer: In response to your request for the inclusion of supporting results from previous studies in our discussion, we have meticulously revised the discussion section to incorporate more substantial evidence and references to prior research. Specifically, we have enriched lines 402-408 and lines 417-425 with additional details, now presented as follows: 

“We acknowledge the pivotal role of previous studies in laying the groundwork for our research on bacteriophage therapy as a viable alternative to traditional antibiotics. These studies have demonstrated the effectiveness of bacteriophages in not only targeting E. faecalis but also in providing a basis for their application in treating various antibiotic-resistant bacterial infections [27, 28]. The selection of E. faecalis as our primary focus was driven by its alarming resistance to a wide array of antibiotics, including vancomycin, and its significant impact on healthcare systems worldwide.”

“The bacteriophage SAM-E.f 12 demonstrated remarkable effectiveness against select isolates of clinical E. faecalis, evidenced by the formation of clear plaques in 70% of the 60 strains tested in the spot assay. This notable specificity, underscored by the lack of lytic activity against E. faecium, highlights SAM-E.f 12's targeted action, especially against vancomycin-resistant E. faecalis strains. Such specificity is crucial for addressing the growing challenge of antibiotic resistance and aligns with the findings of previous research, which have documented the efficacy of specific phages in selectively targeting E. faecalis. This alignment not only corroborates SAM-E.f 12's potential in phage therapy but also reinforces the importance of phage specificity in the development of targeted antibacterial treatments [35, 36].”

Reviewer comment 3: 

• Why you only choose Enterobacter faecalis instead of E. coli? Is there any special characteristics having Enterobacter faecalis? Explain clearly in discussion part.

Response to Reviewer Comment 3:

To clarify our rationale for selecting Enterococcus faecalis over E. coli, we have thoroughly revised the initial part of the discussion section, specifically lines 383-393, to elucidate our objectives and decision-making process. The revised passage now reads:

“E. faecalis, a Gram-positive bacterium, poses a formidable challenge in clinical settings due to its intrinsic resistance to a broad spectrum of antibiotics, including those vital for combatting bacterial infections [1, 2, 25]. This resistance has prompted its inclusion in the World Health Organization's priority pathogen list [26]. This resistance has not only placed E. faecalis on the World Health Organization's priority pathogen list but also highlighted the critical need for innovative therapeutic strategies to combat such resilient bacterial strains. Particularly noteworthy is E. faecalis's resistance to antibiotics like vancomycin, accentuating the exigency to explore alternative treatment modalities [27]. Phage therapy, characterized by a targeted approach, emerges as an advantageous strategy, especially in combating antibiotic-resistant E. faecalis strains, aligning with broader efforts to address antibiotic resistance through innovative antimicrobial exploration [27, 28].”

The overall paper is good. I strongly agree this work as a publication.

Closing Remarks:

We sincerely thank you for your positivity and constructive comments. We believe that these revisions have significantly strengthened our manuscript, making a compelling case for the publication of our work. We look forward to the possibility of our revised manuscript being favorably considered for publication.

Warm regards

---

## [Editor Report · Decision Letter 1]

14 Mar 2024

Insights into the Novel Enterococcus faecalis Phage: A Comprehensive Genome Analysis

PONE-D-24-03766R1

Dear Dr. Shafiei,

We’re pleased to inform you that your manuscript has been judged scientifically suitable for publication and will be formally accepted for publication once it meets all outstanding technical requirements.

Kind regards,

Vinod Kumar Yata, PhD

Academic Editor

PLOS ONE

---

## [Editor Report · Acceptance letter]

25 Apr 2024

PONE-D-24-03766R1 

PLOS ONE

Dear Dr. Shafiei, 

I'm pleased to inform you that your manuscript has been deemed suitable for publication in PLOS ONE. Congratulations! Your manuscript is now being handed over to our production team.

Kind regards, 

on behalf of

Dr. Vinod Kumar Yata 

Academic Editor

PLOS ONE